# What Doesn't Kill You Makes You Robust(er): How to Adversarially Train against Data Poisoning

## Abstract

Data poisoning is a threat model in which a malicious actor tampers with training data to manipulate outcomes at inference time. A variety of defenses against this threat model have been proposed, but each suffers from at least one of the following flaws: they are easily overcome by adaptive attacks, they severely reduce testing performance, or they cannot generalize to diverse data poisoning threat models. Adversarial training, and its variants, are currently considered the only empirically strong defense against (inference-time) adversarial attacks. In this work, we extend the adversarial training framework to defend against (training-time) data poisoning, including targeted and backdoor attacks. Our method desensitizes networks to the effects of such attacks by creating poisons during training and injecting them into training batches. We show that this defense withstands adaptive attacks, generalizes to diverse threat models, and incurs a better performance trade-off than previous defenses.

## 1 Introduction

As machine learning systems consume more and more data, the data curation process is increasingly automated and reliant on data from untrusted sources. Breakthroughs in image classification (Russakovsky et al., 2015) as well as language processing (Brown et al., 2020) are built on large corpora of data *scraped* from the internet. Automated scraping, in which data is collected directly from online sources, leaves practitioners vulnerable to *data poisoning* in which bad actors tamper with the data so that models trained on this data perform poorly or contain *backdoors* embedded in them (Gu et al., 2019; Shafahi et al., 2018). These attacks present security vulnerabilities that persist even if the data is labeled and checked by crowd-sourced human supervision. In essence, entire machine learning pipelines can be compromised if the input data is modified maliciously - even if the modification appears minor and inconspicuous to a human observer. This mounting threat has instilled fear especially in industry practitioners whose business models rely on powerful neural networks trained on massive volumes of scraped data (Kumar et al., 2020).

In response to this growing threat, recent works have proposed a number of defenses against data poisoning attacks (Li et al., 2021b; Goldblum et al., 2020). Existing defense strategies suffer from up to three primary shortcomings:

1. In exchange for robustness, they trade off test accuracy to a degree that is intolerable to real-world practitioners (Geiping et al., 2021).

2. They are only robust to specific threat models but not to adaptive attacks specially designed to circumvent the defense (Koh et al., 2018; Tan & Shokri, 2020).

3. They apply only to a specific threat model and do not lend a generally applicable framework to practitioners (Wang et al., 2019), that could extend to novel attacks.

We instead propose a variant of *adversarial training* that harnesses adversarially poisoned data in the place of (test-time) adversarial examples. We show that this strategy exhibits both an improved robustness-accuracy trade-off as well as greater flexibility for defending against a wide range of threats including adaptive attacks.

Adversarial training desensitizes neural networks to test-time adversarial perturbations by augmenting the training data with on-the-fly crafted adversarial examples (Madry et al., 2018). Similarly, we

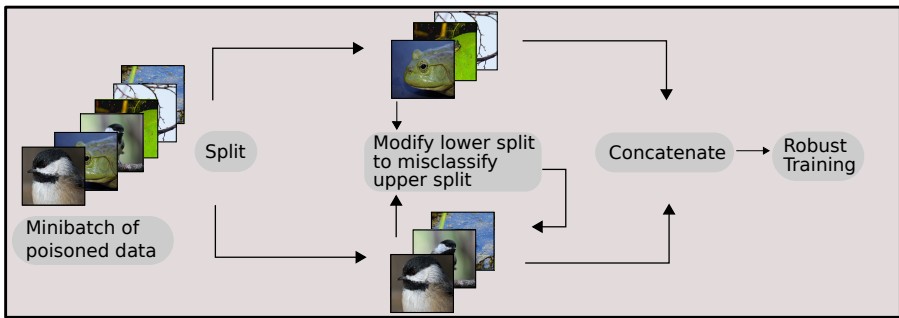

Figure 1: Data poisoning attacks require a new approach to adversarial training to robustify machine learning models against this threat model.

modify training data in order to desensitize neural networks to the types of perturbations caused by data poisoning - yet adapting this robust training framework to data poisoning requires special consideration of this new threat model. In contrast to contemporary work on training against data poisoning such as Li et al. (2021a) the defense dynamically trains a robust model without the need to identify poisoned data points and provide benefits by adapting specifically to targeted data poisoning instead of using default adversarial training as in Tao et al. (2021) or poisoned data generated offline as in Radiya-Dixit & Tramer (2021). We demonstrate the effectiveness of this framework at defending against a range of data tampering threat models across several training regimes and visualize the impact of the defense in feature space. We further compare the proposed defenses to a range of related defense strategies.

## 2 RELATED WORK

Data poisoning is a class of threat scenarios focused on malicious modifications to the training data of a machine learning model. See Goldblum et al. (2020) for an overview of dataset security. Data poisoning attacks can either focus on reducing the overall model performance or on backdoor attacks that introduce malicious behavior which is triggered by a specific visual pattern or target image, thus breaking model *integrity* (Barreno et al., 2010).

In this work, we focus on attacks against model integrity which can be further distinguished by the nature of their trigger mechanism. In *backdoor trigger attacks* (Gu et al., 2019; Turner et al., 2018), the attack is triggered by a specific backdoor pattern or patch that can be added to target images at test time, whereas *targeted data poisoning* (Shafahi et al., 2018; Zhu et al., 2019) is triggered by a predefined target image. In contrast to targeted poisoning, backdoor trigger attacks can be applied to multiple target images but require target modifications to be active during inference, while targeted attacks are activated by specific, but unmodified targets.

### 2.1 DATA POISONING ATTACKS:

Attacks can be further categorized by the precise training setup they anticipate their victims to employ. Some attacks assume that the victim will only fine-tune their model on the poisoned data or will train a linear classifier on top of a pre-trained feature extractor (Saha et al., 2020; Zhu et al., 2019; Shafahi et al., 2018). These methods are effective against practitioners engaging in transfer learning with pre-trained models like those found in popular repositories such as Paszke et al. (2017). Other attacks work even when the victim trains their model from scratch on the poisoned data (Huang et al., 2020; Geiping et al., 2021). Generally, the simpler the victim's training procedure, the more readily the attacker can anticipate the effect of their perturbations and thus, the easier the attacker's job. Here, we briefly detail prominent attacks against which we test our defense:

**Feature Collision:** Shafahi et al. (2018) present a targeted poisoning attack that perturbs training data so that its deep features collide with those corresponding to a target image. At the same time, the attack penalizes the size of perturbations applied to training data in order to maintain visual similarity between the original and perturbed images.

Aghakhani et al. (2020), as opposed to colliding images in feature space, surround target images in feature space to increase the success of feature collision. This method improves on the work of Zhu et al. (2019), which constructs a convex polytope around the target based on an ensemble of models.

**Bilevel Optimization:** MetaPoison (Huang et al., 2020) generates poisoned data based on unrolling the bilevel objective encountered in targeted data poisoning for several steps and optimizing the unrolled objective over an ensemble of models at different stages in training, leading to an attack that is robust to new initializations in from-scratch training as well as to changes in model architectures.

**Gradient Matching:** Witches' Brew (Geiping et al., 2021) instead approximates the bilevel objective by gradient matching, leading to a computationally efficient attack (in comparison to MetaPoison (Huang et al., 2020)), combining the efficiency of feature collision attacks with the success of bilevel approximations. This attack was already adapted to be effective against data augmentation and differential privacy, showing the need for strong defenses against adaptive attacks.

**Hidden Trigger Backdoor Attacks:** Saha et al. (2020) present a backdoor trigger attack wherein the attacker modifies training data within an $\ell^{\infty}$ bound to cause a vulnerability for a pre-selected $\ell_0$ patch added to target validation images. This way, the attack is generalize from a chosen target to only a chosen patch with can be added to any data.

## 2.2 DEFENSES AGAINST DATA POISONING:

Defenses can be broadly classified into *filter defenses* which attempt to detect and remove or sanitize malicious training data, *robust training* algorithms which use a training routine that yields robust models even on malicious training data, and *model repair* methods that train models on poisoned data and attempt to repair the poisoned models after training. Filter defenses are easy to deploy as they simply add a pre-processing step, but they require extensive hyperparameter tuning and rely on the assumption that only a small fraction of the dataset is poisoned. Furthermore, filter defenses can often be overcome by adaptive attacks (Koh et al., 2018; Tan & Shokri, 2020). Any defense reduces model performance (in terms of validation accuracy); filter defenses reduce performance as a result of training the new model on fewer samples, robust training methods do so by deviating from standard training practices which are tuned for accuracy in order to increase robustness, and model repair methods harm accuracy by pruning away potentially important neurons.

Numerous options have been proposed for detecting poisoned data. Tran et al. (2018a) detect *spectral signatures* associated with backdoor triggers based on their correlation with the top right singular vector of the covariance matrix formed by feature representations, and additional detection scores can be found in Paudice et al. (2018). Peri et al. (2020) detect poisoned data by clustering based on *deep KNN*, re-labeling data based on the nearest neighbors in feature space. Chen et al. (2019) cluster training data based on activation patterns in feature space. Another method detects images containing backdoor triggers by flagging images whose corresponding predictions do not change when they are combined with clean samples (Gao et al., 2019). Yet, any measure of "anomaly" that is used to filter images can also be used to generate poisoned data which minimizes the anomalous property, thus adaptively defeating the defense (Koh et al., 2018).

Robust training algorithms may incorporate strong data augmentations (Borgnia et al., 2020), randomized smoothing (Weber et al., 2020), or may partition data into disjoint pieces and train individual models on the partitions, performing classification via majority voting at test-time (Levine & Feizi, 2021). Another popular robust training strategy harnesses differentially private SGD (Abadi et al., 2016; Ma et al., 2019). Hong et al. (2020) note that the addition of noise in differentially private SGD is the primary factor controlling robustness to poisoning. However, differential privacy is an extreme and general definition of robustness to data manipulations, compared to robustness specific to data poisoning. This strategy consequently incurs a significant performance penalty (Jayaraman & Evans, 2019), and these algorithms can even be adaptively attacked by modifying gradient signals during poison generation in the same manner as in the defense (Veldanda & Garg, 2020). Other robust training schemes are proposed in Li et al. (2021a); Tao et al. (2021); Radiya-Dixit & Tramer (2021). Yet, adaptive attacks can bypass these defenses by creating poisoned data whose activation patterns mimic those of clean data. In order to counteract the loss of performance induced by pruning, some methods fine-tune the pruned model on clean data (Liu et al., 2018; Chen et al., 2020). But this process is only effective when the defender possesses large quantities of trustworthy clean data.

# 3 UNDERSTANDING ADVERSARIAL TRAINING FOR DATA POISONING

Adversarial training (Madry et al., 2018; Sinha et al., 2018) reduces the impact of test-time adversarial attacks and is generally considered the only strong defense against adversarial examples. Adversarial training solves the saddle-point problem,

$$\min_{\theta} \mathbb{E}_{(x,y)\sim\mathbb{D}} \left[ \max_{\Delta \in S} \mathcal{L}_{\theta}(x + \Delta, y) \right], \tag{1}$$

where $\mathcal{L}_{\theta}$ denotes the loss function of a model with parameters $\theta$, and the adversary perturbs inputs $x$ from a data distribution $\mathbb{D}$, subject to the constraint that perturbation $\Delta$ is in $S$. Peri et al. (2020) notes that adversarial training against test-time evasion attacks already confers a degree of robustness against data poisoning at a performance cost. Our proposed strategy is an adaptation of adversarial training to poisoning, resulting in a stronger defense that degrades performance less than differentially private SGD or adversarial training against evasion attacks. The capabilities of an attacker depend on its knowledge of the defender's training setup, so we now enumerate a series of assumptions concerning the knowledge of the attacker and defender before presenting our framework in precise detail.

**Preparing for a strong threat model.** In order to harden the model against a wide range of poisons, we train against a strong surrogate attacker. The differences between the surrogate threat model and that of a real-world attacker concern the attacker's access to the defender's training routine. The surrogate attacker in our training algorithm is aware of the defender's training protocol (e.g. learning rate, optimization algorithm), architecture, and defense strategy but can neither influence training nor intercept random factors such as initialization and mini-batch sampling. In cases where the defender only re-trains a component of a model or fine-tunes the model, the exact baseline pre-trained model, including its parameters, is known to both parties. The attacker's trigger (a target image for targeted poisoning or a specific patch for backdoor attacks) is unknown to the defender, and we do not assume that the defender possesses additional, vetted, clean data. In order to constrain the attacker, the defender chooses a $|| \cdot ||$-norm perturbation budget $\varepsilon$ against which they seek robustness.

Since the attacker possesses such strong knowledge concerning the defender's training routine, the threat it poses constitutes a near worst-case analysis. If more factors, such as model definition or parts of the training protocol, are hidden from the attacker, then the quality of the defense can only improve. On the other hand, the defender needs to set a $\varepsilon$ bound within which to be robust against attacks - if there were no such limit, then the attacker could arbitrarily modify data.

## 3.1 ADVERSARIALLY TRAINING POISON IMMUNITY

In contrast to adversarial attacks at test-time, the objective for *targeted data poisoning* is itself already a bilevel objective. For a given target $x_t$ with an intended adversarial label $y_t$, targeted data poisoning optimizes poisoned data points $x_p$, so that models trained with them exhibit low loss on the adversarial label $y_t$ instead of its original label $y_o$:

$$\min_{x_p} \mathcal{L}(x_t, y_t, \theta(x_p)) \qquad \text{s.t. } \theta(x) = \arg\min_{\theta} \sum_{i=1}^{N} \mathcal{L}(x_i, y_i, \theta). \tag{2}$$

The optimal defense against this attack then develops into a two-player game between attacker and defender. While the attacker minimizes the loss on the target, the defender maximizes it with ($\theta(x)$ as defined above),

$$\min_{x_p} \max_{\Delta \in S} \mathcal{L}(x_t, y_t, \theta(x_p + \Delta)). \tag{3}$$

However, this formulation reveals the central trick of adversarially training against data poisoning: Both attacker and defender modify the same variable $x = x_p + \Delta$. This means that any known algorithm used to approximate the bilevel poisoning objective for an attack (e.g. Huang et al. (2020) or Geiping et al. (2021)), is valid as an approximation for a defense.

In the formulation above, the solution for the defender is simple, the defender can optimize the poisoning objective Eq. (2) for $x_p$ and then set $\Delta = -x_p$. Yet, in practice, the defender has neither knowledge of the specific data point $x_t$ targeted by the attack, nor can change their strategy in

response to the attacker. The optimal choice is hence to sample surrogate targets $(x_t, y_t)$ from the data distribution, optimize $x_p$, and take steps to maximize $\mathcal{L}(x_t, y_t, \theta)$:

$$\max_{\Delta \in S} \mathbb{E}_{(x_t, y_t) \in \mathcal{D}} \left[ \min_{x_p} \mathcal{L}(x_t, y_t, \theta(x_p + \Delta)) \right]. \tag{4}$$

Instead of maximizing the adversarial loss, the defender can equivalently minimize the loss of the true label for the sampled target, i.e. $\mathcal{L}(x_t, y_o, \theta)$. To implement the objective Eq. (4), we sample mini-batches of data and then first split these batches into two subsets of data, $(x_p, y_p)$ and $(x_t, y_t)$, with probability $s$ for a data point to be placed in the first ("poison") split. For the sampled target $x_t$ we then approximately minimize $\mathcal{L}(x_t, y_t, \theta(x_p + \Delta))$ through a known data poisoning attack, which yields $x_p$. We can then update the model parameters to minimize $\mathcal{L}(x_t, y_o, \theta)$ and $\mathcal{L}(x_p, y_p, \theta)$, i.e. we train the model on the concatenated output of surrogate poisons and targets, as seen in Fig. 1. This way we alternate between both steps in Eq. (4) effectively.

Interestingly, contemporary work in Li et al. (2021a) instead maximizes adversarial loss in the outer objective and instead of optimizing in expectation detects and maximizes over detected poisoned data points $x_p$, leading to a different approximation to Eq. (3), whereas work in Tao et al. (2021) focuses on adversarial training against attacks on model availability, which can be understood by replacing Eq. (2) by a maximization of loss over all data points, leading to the min-max structure of Eq. (1) as a defense. Work in Radiya-Dixit & Tramer (2021) instead proposes to optimize Eq. (4) by first generating additional poisoned data $x_p$ for a fixed pretrained model and then training with all data, which corresponds to approximating the objective Eq. (4) in an offline fashion.

These considerations hold not only for targeted data poisoning, but likewise apply to poisoning with other modalities, such as backdoor triggers. In those cases, both $x_p$ and $x_t$ are optimized (or sampled) instead of only $x_p$.

**Example: Defending against Gradient Matching.** While our methodology can be applied to any data poisoning attack, there are several considerations to make when adapting the attack into a format that is applicable and practical to run in each mini-batch of training. We detail these considerations for a recent attack (Geiping et al., 2021).

The cosine similarity objective of the attack is originally evaluated on a clean surrogate model trained by the attacker and the attack is optimized for a significant number of iterations ($n = 250$ in the original work). To apply it during training, we first replace the clean model used in the attack with the current model in the current state of training - this is actually an advantage for the defender. While the attacker needs to create poisons on a surrogate model, the defender can use the exact model, making it easier to create effective poisons. Secondly, similar to adversarial training, the number of attack iterations can be reduced. In practice, we choose $n = 5$ during the defense, as a compromise between creating a strong attack and spending a limited time budget, as the attack is naturally stronger due to its basis in the current state of training. Third, we need to choose malicious labels $y_t$. These labels could be chosen entirely at random, however then the average gradient over all targets would likely be small. However, poisoned data points in Geiping et al. (2021) are in practice chosen from the same class as the target adversarial label, and this choice can be replicated for the randomly chosen subset of poisoned data points $x_p$ with labels $y_p$ by choosing $y_t$ as the label that appears most often in $y_p$.

**Other defenses can be viewed as special cases of poison immunity.** This methodology generalizes and explains previous work on defenses against poisoning. In Borgnia et al. (2020), strong data augmentations such as mixup (Zhang et al., 2018) and cutout (DeVries & Taylor, 2017) are proposed as defenses against data poisoning. These defenses are special cases of the proposed poison immunity; when Algorithm 1 is used to defend against a watermarking attack (which superimposes the target data onto poison data with low opacity), then the attack is equivalent to mixup data augmentation with mixing factor $\alpha = 1 - \varepsilon$. Likewise, implementing Algorithm 1 against a patch attack attack reduces to patching randomly selected pairs of data points with a random patch. If this patch is chosen to be uniformly gray, then this defense is exactly equivalent to the cutout data augmentation.

### 3.2 ADAPTIVE ATTACK SCENARIOS

Crucial for the design of new defense algorithms is their ability to withstand adaptive attacks, i.e. attacks that can be modified to respond to a defense algorithm when the attacker is aware of the

---

**Algorithm 1** Modified iterative training routine for poison immunity.

---

**Input:** Split probability $s \in (0, 1)$.
**repeat**
    Sample mini-batch of data $\{x_i, y_i\}_{i=1}^n$,
    Split data randomly into two subsets $x_p, x_t$ with probability $s$
    Draw malicious labels $y_t$ for $x_t$
    Apply a data poisoning attack to minimize $\mathcal{L}(x_t, y_t, \theta(x_p))$ via $x_p$
    Concatenate $x_p, x_t$ into a new batch $x_m$ with unchanged labels $\{y_i\}_{i=1}^n$
    Update model based on new data $x_m$
**until** training finished

---

defense. While this principle has been well-regarded in literature about adversarial attacks at test-time Carlini et al. (2019); Tramer et al. (2020), it has not been applied as rigorously for data poisoning.

The defense proposed in this work is exceedingly effective against non-adaptive models (evaluating the exemplary case of gradient matching), as the difference in training regimes leads to incorrect perturbations computed by the attacker that relies on a pre-trained surrogate model. However, this would also be the case for most modifications to the training procedure, such as adding data augmentations or changing learning rates or optimizer settings. As such, we find that the optimal way to attack this defense is for the attacker to re-train their pre-trained model with exactly the same defense and the same hyperparameters. The attacker can then more accurately estimate the target gradient (for gradient matching) or target features (for feature collision). We also investigated the possibility of applying Algorithm 1 during the optimization of poisoned data itself as an additional stochastic input modification. However, this modification weakens the attack by gradient masking, making it too difficult for the attacker to optimize the poisoned data. This behavior mirrors (test-time) adversarial attacks, where it is non-optimal to add additional perturbations during the creation of an adversarial perturbation. As we will find in the next section, the defense has a major impact on the feature space of a model, which may make it difficult to bypass it with better adaptive attacks.

## 4 ANALYSIS

To understand the effect of the proposed poison immunity scheme qualitatively, we conduct an analysis of feature space visualizations for several attacks. Shafahi et al. (2018), who introduced feature collision attacks, illustrate their poisoning method by visualizing the feature space collisions between poisoned data and the target. These experiments are carried out in the transfer setting, where the feature representation of the model is fixed and known to both parties. We run Bullseye Polytope, a recent and improved feature collision attack (Aghakhani et al., 2020), in a strong attack setting of 500 poisoned examples for a ResNet-18 pre-trained on CIFAR-10 and re-trained on the poisoned dataset. We visualize the feature space by plotting the projections of feature vectors of data on to the vector connecting the centroids of the poison (base) class and the target class and it's orthogonal (generated using PCA) in the $x$-$y$ plane and the softmax output for poison (base) class for each of the points on the $z$ axis. This way we expect to see both classes to form separate clusters in feature space ($x$-$y$ plane) and to be further separated by the poison class probability, which is low for images from the target class and high for images from the poison class.

Fig. 2a shows the effects of an attack on a baseline model. The poisoned images (red) move to *collide* with the target (black triangle) in feature space as seen by their overlap in the $x$-$y$ plane, while maintaining their original label ($z$-axis), subsequently leading to a misclassification of the target image as image from the poison class - this is a *feature collision* attack. Fig. 2b however contrasts this collision with the effects of poisons on a defended model. Two effects stand out: First, the poisons, which were optimized to collide with the target, no longer cluster around the target (refer also to the 2D visualization in Fig. 12), indicating that straightforward collisions are difficult to achieve against the robust model. Second, poisoned images close to the target are now predicted as the target base class shown by their descend on the $z$-axis; While we would expect the first effect for any defense, the second effect really breaks the attack. the defended model is robust enough to assign poisoned images a label that agrees with their feature representation, even though this assignment contradicts the given labels of these images. In essence, the poisoned images are treated like images from the

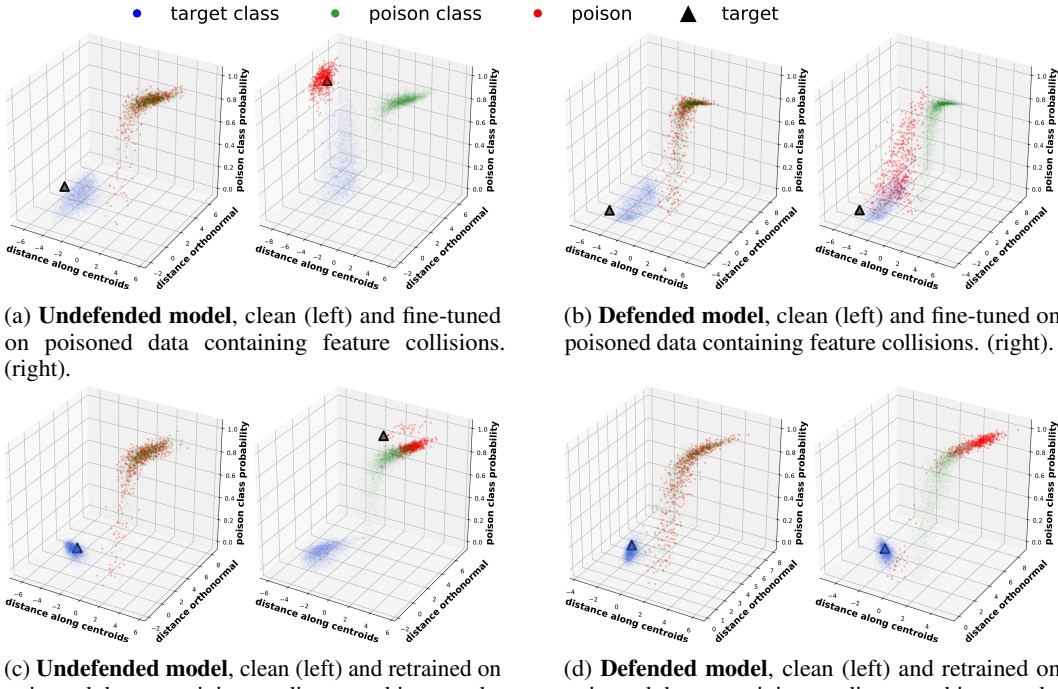

(a) **Undefended model**, clean (left) and fine-tuned on poisoned data containing feature collisions. (right).

(b) **Defended model**, clean (left) and fine-tuned on poisoned data containing feature collisions. (right).

(c) **Undefended model**, clean (left) and retrained on poisoned data containing gradient matching attacks (right).

(d) **Defended model**, clean (left) and retrained on poisoned data containing gradient matching attacks (right).

Figure 2: Visualization of the effects of poisoning attacks against an undefended and a defended model. Top: Feature collisions via Aghakhani et al. (2020). Bottom: Gradient matching as in Geiping et al. (2021). The target image is marked by a black triangle and is originally part of the class colored blue. The poisoned images are colored red and are part of the class colored green. The $x$-$y$ axis in each diagram corresponds to a projection of the principal direction separating both classes, while the confidence in the original target class is marked on the $z$-axis. The defended models generate a feature space in which poisons (red) behave consistently with the robust model, prevening collsions in Fig. 2b and alignment in Fig. 2d, so that the target (black) remains correct.

target class but with a noisy label. This property reverses the attack. Instead of moving the target into the poison class, the poisoned images are drawn into the target class as it matches their feature representation. The model stays consistent and is able to defend against the strong attack analyzed here.

In addition to feature collision attacks, in Fig. 2, we analyze the defense against the gradient matching attack of Geiping et al. (2021) in the from-scratch setting, where the model is fully re-trained. The attack can be seen to be effective in Fig. 2c, changing the decision boundary of the model to fit the target without collisions by clustering poisons opposite to the target in feature space, significantly moving the target. However, this is prevented by the defense as seen in Fig. 2d. The robust model is not modified by the clustering of poisoned images, and outliers seen in the undefended model are again reclassified as the target class leading to a consistent decision. An interesting side effect of the defense for both attacked and clean models is that the model itself is generally less over-confident in its clean predictions. We compute similar outcomes also for other attacks, such as patch attacks, which we show in the appendix in Fig. 6 to Fig. 15.

## 5 EXPERIMENTS

This section details a quantitative analysis of the proposed defense for the application of image classification with deep neural networks. To fairly evaluate all attacks and defenses, especially in light of Schwarzschild et al. (2020) discussing the difficulty in comparing attacks across different evaluation settings, we implement all attacks and defenses in a single unified framework, which we will make publicly available. For all experiments, we measure *avg. poison success* over 20 trials, where each trial represents a randomly-chosen attack trigger from a random class and a separately

Table 1: Quantitative result for several attacks and their defense by poison immunity with $s = 0.75$, showing avg. poison success with standard error (where all trials have equal outcomes, we report the worst-case error estimate $5.59\%$). Additional details about each attack threat model can be found in the Appendix E.2. The proposed defense significantly decreases success rates over a wide range of attacks and scenarios without any hyperparameter changes. This table evaluates gradient matching (GM) with both squared error (SE) and cosine similarity (CS). The evaluation column with the strongest attack uses gradient matching with cosine similarity.

| Attack | Scenario | Undefended | Defended | |
| --- | --- | --- | --- | --- |
| | | | Same Attack | Strongest Attack |
| MetaPoison | From-scratch | 69.50% (±9.34) | 10.00% (±9.49) | 10.00% (±9.49) |
| Gradient Matching (CS) | From-scratch | 90.00% (±6.71) | 0.00% (±5.59) | 0.00% (±5.59) |
| Bullseye Polytope | Fine-tuning | 80.00% (±8.94) | 0.00% (±5.59) | 8.33% (±7.98) |
| Bullseye Polytope | Transfer | 100.00% (±5.59) | 10.00% (±6.71) | 0.00% (±5.59) |
| Poison Frogs | Transfer | 100.00% (±5.59) | 15.00% (±7.98) | 0.00% (±5.59) |
| Gradient Matching (SE) | Transfer | 95.00% (±4.87) | 0.00% (±5.59) | 5.00% (±4.87) |
| Hidden Trigger Backdoor | Transfer | 55.59% (±5.65) | 24.78% (±6.82) | 3.32% (±0.79) |

attacked and trained model. The sampling of randomized attack triggers is crucial to estimate the average performance of poisoning attacks, which are generally more effective for related class labels. We discuss additional experimental details in Appendix E.

**Defending against $\ell^\infty$ threat models** We focus on defending against threat models, in which the attacker may modify some percentage of training data within an $\ell^\infty$ bound, i.e. change every pixel slightly. This covers all mentioned targeted data poisoning attacks as well as hidden trigger backdoor attacks (Saha et al., 2020).

To evaluate the proposed defense mechanism thoroughly, we consider a variety of attacks in different scenarios and distinguish three scenarios with increased difficulty for the attacker, *transfer* where the defender only re-trains the last linear layer of a model, *fine-tuning*, where the defender re-trains all layers, and *from-scratch* where the defender trains a completely new model.

We first apply the proposed defense against a range of attacks and settings in Table 1, choosing $s = 0.75$ for all targeted data poisoning attacks and no additional modifications. All attacks shown are adaptive, if possible. In the fine-tuning and transfer scenarios, the pre-trained model is defended but known to the attacker exactly. In all cases, we observe that while the attacks are highly effective against an undefended model, our defense steeply reduces their effectiveness. These encouraging results suggest that the proposed methodology is a strong strategy that can be robustly applied across a range of attacks and may also generalize to future attacks.

A natural question to ask is whether this defense, which trains against one specific surrogate attack, can be circumvented when the real attacker utilizes a different attack. Surprisingly, we find in Table 1 that using gradient matching, a strong attack, as surrogate during training successfully defends against a range of other attacks. This corroborates findings in Madry et al. (2017) where it was shown that training with the strongest test-time adversarial attack, also defends against weaker test-time attacks - an mechanism which fortunately appears to also hold when defending against data poisoning.

**Comparison to Other Defenses** In this subsection, we compare the proposed defense to other existing defense strategies against data poisoning including differentially private SGD, adversarial training, various data augmentations, and filter defenses. For differentially private SGD and adversarial training, we test several noise levels and perturbation budgets, respectively. When comparing to filtering defenses, we allow an optimal hyperparameter choice by supplying the exact number of poisons in the training set, although this information would be unknown in practice. We analyze poison immunity training with varying levels of $s$ to show the trade-off of performance and security. We test the gradient matching attack proposed in Geiping et al. (2021), for a ResNet-18 trained on CIFAR-10 with budget $1\%$ and $\varepsilon = 16$ (the same setting as proposed in that work). While previous

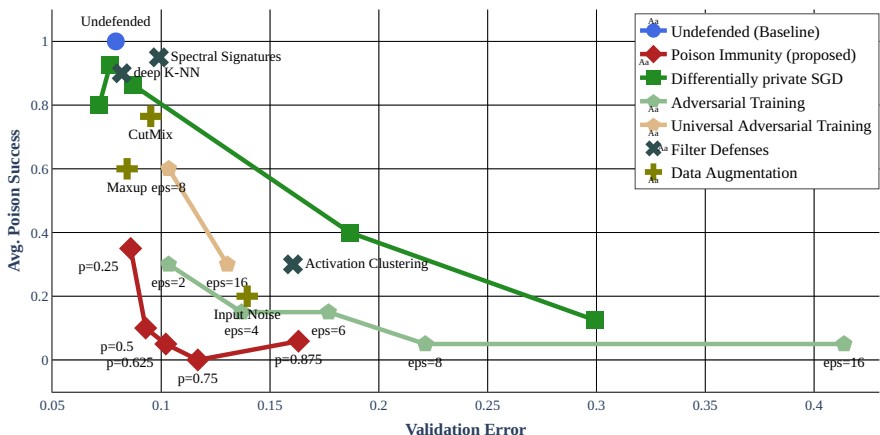

Figure 3: Avg. Poison Success versus validation accuracy for various defenses against the gradient matching attack of (Geiping et al., 2021) in the from-scratch setting. The baseline undefended model is shown in blue, the proposed defense in red. The differentially private SGD is shown for noise values from 0.0001 to 0.01. The proposed defense is a strong trade-off of robustness and accuracy.

defenses were shown to be ineffective in Geiping et al. (2021), we now show in Fig. 3 that the proposed poison immunity defense is an extremely effective defense in the from-scratch setting, yielding a much stronger protection than filter defenses, but with only mild trade-off in validation accuracy compared to differential privacy and classical adversarial training.

**Releasing Robust Models** So far, we have considered scenarios in which the defense described in Algorithm 1 is always active, even during the fine-tuning procedure in the transfer setting. However, especially in the transfer setting, we are interested in the inherent robustness of models and its transferability. We thus analyze a setting in which the base model is trained robustly via poison immunity, but the last layer is re-trained non-robustly on poisoned data. For the bullseye polytope attack in the transfer setting, this approach leads to an avg. poison success of only $20.00\%(\pm 8.94)$ and validation accuracy of $88.66$ when using a base model trained robustly with poison immunity via $s = 0.75$, compared to the best defense of $10.00\%(\pm 6.71)$ - a significant part of the overall robustness is already encoded into the base model. The inset figure visualizes that the target confidence remain consistent, even if the fine-tuning is nonrobust.

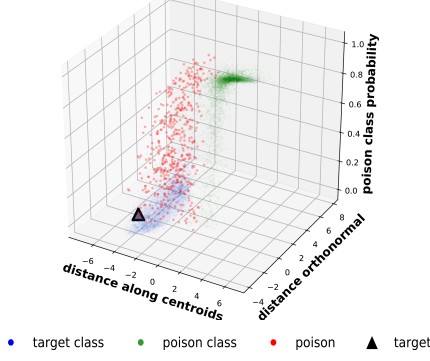

Figure 4: Feature Defense. A robust base model can withstand feature collision attacks even when fine-tuning nonrobustly on poisoned data. Note how *no* collision forms around target and the model remains robust.

## 6 CONCLUSIONS

In this work, we adapt adversarial training to defend against data poisoning attacks. In addition to demonstrating the strong defensive capabilities of our method, poison immunity, we analyze the feature space of defended models and observe mechanisms of defense. We further evaluate the proposed defense against a variety of attacks on deep neural networks for image classification, successfully adapting to and defending against feature collisions, gradient matching, and hidden trigger backdoors. We stress that we believe this strategy to be a general paradigm for defending against data tampering attacks that can extend to novel future attacks.

ETHICS STATEMENT

Data poisoning attacks have the potential to disrupt machine learning pipelines and hinder data collection and curation. Especially when collected data is of unknown quality and possibly contaminated with poisoned samples, we hope that defense strategies like we propose can be useful in mitigating harmful effects.

REPRODUCIBILITY STATEMENT

We refer to the appendix for details for all attacks and defenses evaluated in this work and regarding the experimental setup. We further supply code that allows for the reproduction of all experiments in a consistent framework.

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

## A  DEFENDING AGAINST $\ell^0$ THREAT MODELS

In this section we further investigate patch attacks as in Gu et al. (2019) who trigger backdoors by adding an $\ell_0$ bounded patch to training images in a given class, causing a network trained on these images to associate the patch with the given class. Then, the attacker patches test-time images (of a different class) with the same patch in the hopes that the network mis-classifies the images. Most backdoor attacks fall under an $\ell^0$ threat model, or more specifically a limited $\ell^0$ norm for connected rectangular region, a patch. Defending against such attacks through adversarial training techniques is not as well-understood, even for test-time adversarial attacks (Rao et al., 2020), as $\ell^\infty$ threat models. Although Eq. (4) remains valid, finding the $\arg\min$, i.e. the worst-case patch to apply, appears difficult from an optimization perspective.

We evaluate this threat model for the example of a challenging backdoor trigger attack in Table 2, where we insert a 4x4 trigger patch into $5\%$ of the CIFAR-10 training data. This patch either is a noisy checkerboard, and as such out-of-distribution for CIFAR-10, or a firefox logo, i.e. a potentially in-distribution semantic feature. The evaluated filtering defenses do not defend well against both patches, whereas differentially private SGD provides some protection against the noise patch. When comparing different instances of the proposed poison immunity strategy, we can compare several strategies to approximate Eq. (4): We can sample randomized patches of varying sizes ([large] noise patch). We can optimize for a worst-case patch content at a random position (optimized patch). We can sample a randomized, but in-distribution, patch (image patch) by sampling a patch from another image in the batch, which is equivalent to CutMix (Yun et al., 2019). The results show that optimizing the worst-case patch fails to significantly improve upon a well-chosen sampled patch. For the noisy checkerboard, we reach the best results when like-wise sampling random noise patches during training, although sampling image patches is also competitive. With the semantically meaningful firefox logo however, we find that training with noisy patches actually only incurs robustness to such noisy patterns, and only limited robustness to other patterns. In contrast, the image-based patch can succeed in both settings.

Table 2: Avg. poison success for various defenses against backdoor triggers attacks, for an attack via a 4x4 patch on $5\%$ of training data. Top table: Baseline and defenses via filtering and differential privacy. Bottom table: Variations of adversarial training against poisons. We evaluate each on a noisy checkerboard patch (Noise) and a patch with semantic meaning, a firefox logo (Sem).

| Patch: | Undefended | Filtering Defenses | | | DP-SGD |
|---|---|---|---|---|---|
| | | Spectr. Sign. | deep-KNN | Act. Clust. | |
| Noise | 69.89 ($\pm$4.47) | 68.62 ($\pm$7.97) | 73.34 ($\pm$5.25) | 79.72 ($\pm$6.82) | 50.24 ($\pm$9.41) |
| Sem. | 79.68 ($\pm$4.50) | 83.93 ($\pm$4.55) | 76.96 ($\pm$5.07) | 86.84 ($\pm$3.86) | 94.40 ($\pm$3.53) |

| Patch: | Poison Immunity | | | |
|---|---|---|---|---|
| | Noise Patch | Large Noise Patch | Optimized Patch | Image Patch |
| Noise | 29.77 ($\pm$6.75) | 33.87 ($\pm$5.64) | 62.70 ($\pm$12.28) | 25.80 ($\pm$3.93) |
| Sem. | 66.06 ($\pm$5.17) | 54.45 ($\pm$6.11) | 79.86 ($\pm$6.24) | 25.62 ($\pm$3.85) |

Figure 5: Effectiveness of the Defense against large attack budgets for the case of a defense against gradient matching in the from-scratch setting on a ResNet-18. In this scenario, the attack already succeeds when $1\%$ of the data is poisoned, but the attacker can increase their budget further and poison more of the data. The defense is still effectively reduces poison success even with large levels of poisoned data with a roughly logarithmic scaling. Note that $10\%$ is the limit for this attack on CIFAR-10, for which all data points from the poisoned class are modified.

## B    ABLATION STUDIES

This section contains ablation studies and additional information. We visualize the effect of increased poison budgets in Fig. 5. Table 3 contains information on clean validation accuracy and runtimes for all experiments in Table 1. In the same way Table 4 extends Table 2 with additional information.

## C    VISUALIZATIONS

We repeat the three-dimensional visualizations shown in Fig. 2 in the main work for additional attacks. For easy comparison, we also repeat the figures appearing in the main work. We show feature collisions via Poison Frogs in Fig. 6 and repeat Bulleye Polytope in Fig. 7. We then repeat gradient matching in Fig. 8, in comparison to backdoor trigger in Fig. 9 and gradient matching (SE) in Fig. 10.

All three-dimensional visualizations are also shown in two dimensions, showing Poison Frogs in Fig. 11, Bullseye Polytope in Fig. 12, gradient matching in Fig. 13, backdoor triggers in Fig. 14 and gradient matching (SE) in Fig. 15.

## D    COMPARISON TO OTHER DEFENSES FOR TRANSFER LEARNING

We additionally compare poison immunity to other existing defenses in the transfer setting, where only the last layer is re-trained on poisoned data. We test feature collisions via Bullseye Polytope (Aghakhani et al., 2020), also for a ResNet-18 trained on CIFAR-10 with a budget of $1\%$ poisoned data within an $\ell^\infty$ bound of $\varepsilon = 16$. This is a setting that is ideal for commonly used filtering defenses, as a large number of poisoned data is collided with the target image, which can be detected and filtered, while the setting is difficult for robust training methods, both due to the large perturbations

Table 3: This is the same table as Table 1 in the main body, with additional information of natural validation accuracy and timing for each run. Best viewed on screen. All shown timings are wall-clock time for an NVIDIA GTX-2080ti with 4 assigned CPUs. Some timings are missing (due to machine heterogeneity), but can be inferred from other rows in the same block (i.e. all transfer experiments take roughly the same amount of time for the "Strongest Attack" defense).

| Attack | Scenario | Undefended | | | Defended | | | | | |
| | | | | | Same Attack | | | Strongest Attack | | |
| | | Poison Acc. | Nat.Acc. | Time | Poison Acc. | Nat.Acc. | Time | Poison Acc. | Nat.Acc. | Time |
| MetaPoison | From-scratch | 69.50% ($\pm$9.34) | 86.11% | 0:28:18 | 10.00% ($\pm$9.49) | 81.34% | 2:06:48 | 10.00% ($\pm$9.49) | 78.40% | 1:09:47 |
| Gradient Matching (CS) | From-scratch | 90.00% ($\pm$6.71) | 92.01% | 0:15:48 | 0.00% ($\pm$5.59) | 88.32% | 4:13:51 | 0.00% ($\pm$5.59) | 88.32% | 4:13:51 |
| Bullseye Polytope | Fine-tuning | 80.00% ($\pm$8.94) | 91.93% | 0:16:31 | 0.00% ($\pm$5.59) | 88.49% | 0:59:58 | 8.33% ($\pm$7.98) | 88.14% | 4:20:03 |
| Bullseye Polytope | Transfer | 100.00% ($\pm$5.59) | 91.97% | 0:08:04 | 10.00% ($\pm$6.71) | 88.64% | 0:39:53 | 0.00% ($\pm$5.59) | 90.34% | - |
| Poison Frogs | Transfer | 100.00% ($\pm$5.59) | 91.93% | 0:08:46 | 15.00% ($\pm$7.98) | 88.50% | 0:39:53 | 0.00% ($\pm$5.59) | 90.54% | - |
| Gradient Matching (SE) | Transfer | 95.00% ($\pm$4.87) | 92.02% | 0:08:08 | 0.00% ($\pm$5.59) | 87.68% | 0:41:08 | 5.00% ($\pm$4.87) | 90.62% | - |
| Hidden Trigger Backdoor | Transfer | 55.59% ($\pm$5.65) | 86.07% | - | 24.78% ($\pm$6.82) | 86.44% | - | 3.32% ($\pm$0.79) | 87.94% | 0:40:26 |

Table 4: Avg. poison success for various defenses against backdoor triggers attacks, for an attack via a 4x4 patch on $5\%$ of training data. Extended version of Table 2, but note the flip in rows and columns. Best viewed on screen.

| Patch: | Noise | | | Semantic | | |
| | Poison Acc. | Nat.Acc. | Time | Poison Acc. | Nat.Acc. | Time |
| Undefended | 69.89% ($\pm$4.47) | 91.77% | 0:16:47 | 79.68% ($\pm$4.50) | 91.67% | 0:16:49 |
| Spectral. Sign. | 68.62% ($\pm$7.97) | 77.31% | 0:23:09 | 83.93% ($\pm$4.55) | 77.42% | 0:23:06 |
| Deep-KNN | 73.34% ($\pm$5.25) | 91.27% | 2:46:10 | 76.96% ($\pm$5.07) | 90.97% | 2:45:59 |
| Act. Clustering | 79.72% ($\pm$6.82) | 81.06% | 0:28:34 | 86.84% ($\pm$3.86) | 80.43% | 0:28:36 |
| DP-SGD | 50.24% ($\pm$9.41) | 69.33% | 0:17:47 | 94.40% ($\pm$3.53) | 69.24% | 0:17:43 |
| Noise Patch | 29.77% ($\pm$6.75) | 92.00% | 0:20:20 | 66.06% ($\pm$5.17) | 91.88% | 0:20:21 |
| Large Noise Patch | 33.87% ($\pm$5.64) | 88.91% | 0:19:43 | 54.45% ($\pm$6.11) | 89.02% | 0:19:28 |
| Optimized Patch | 62.70% ($\pm$12.28) | 91.77% | 6:08:03 | 79.86% ($\pm$6.24) | 92.01% | 6:00:42 |
| Image Patch | 25.80% ($\pm$3.93) | 91.41% | 0:16:45 | 25.62% ($\pm$3.85) | 91.15% | 0:16:40 |

Table 5: Additional quantitative result for several attacks and their defense by poison immunity with $s = 0.75$, showing avg. poison success with standard error (where all trials have equal outcomes, we report the worst-case error estimate $5.59\%$). Additional details about each attack threat model can be found in the Appendix E.2.

| Attack | Scenario | Dataset | Model | Undefended | Defended | |
| | | | | | Same Attack | Strongest Attack |
| Gradient Matching | From-Scratch | GTSRB | ResNet18 | 40.00% ($\pm$15.49) | 10.00% ($\pm$9.49) | 10.00% ($\pm$9.49) |
| Bullseye Polytope | Transfer | GTSRB | ResNet18 | 10.00% ($\pm$9.49) | 0.00% ($\pm$5.59) | 0.00% ($\pm$5.59) |

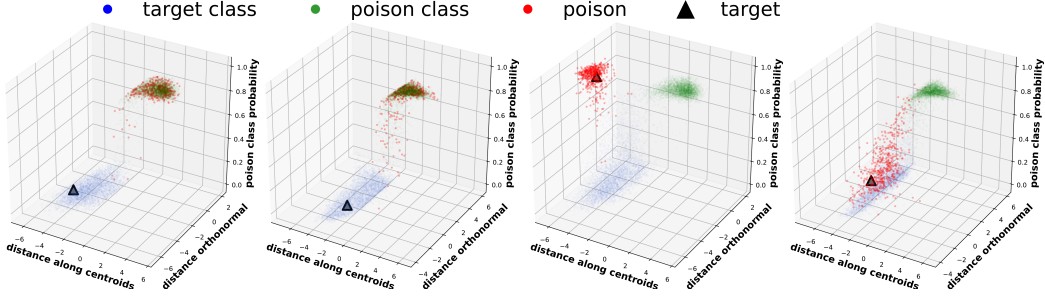

(a) **Undefended model**, clean (left) and fine-tuned on poisoned
data containing feature collisions. (right).

(b) **Defended model**, clean (left) and retrained on poisoned
data containing gradient matching attacks (right).

Figure 6: 3D Visualization of a feature collision attack (via **Poison-Frogs**) against an undefended and a defended model. The defended model significantly hinders feature collisions. The target image is marked by a black triangle and is originally part of the class marked in blue. The poisoned images are marked in red and are part of the class marked in green.

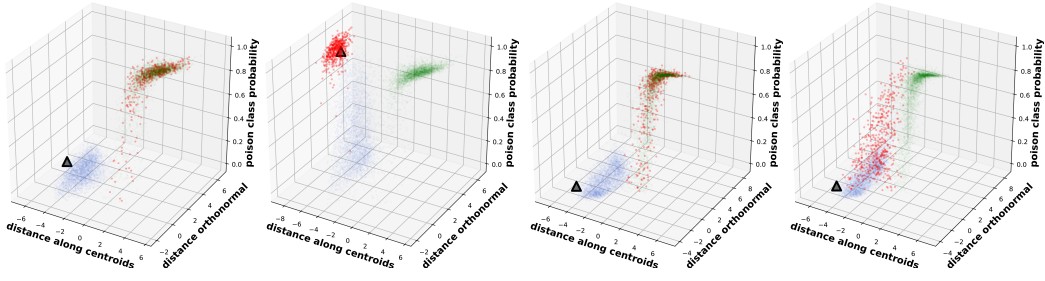

(a) **Undefended model**, clean (left) and fine-tuned on poisoned
data containing feature collisions. (right).

(b) **Defended model**, clean (left) and retrained on poisoned
data containing gradient matching attacks (right).

Figure 7: 3D Visualization of the effects of a feature collision (**Bullseye Polytope**) attack against an undefended and a defended model. The defended model significantly hinders feature collisions. The target image is marked by a black triangle and is originally part of the class marked in blue. The poisoned images are marked in red and are part of the class marked in green. Notably the strong collision seen in the baseline is inhibited by the defense.

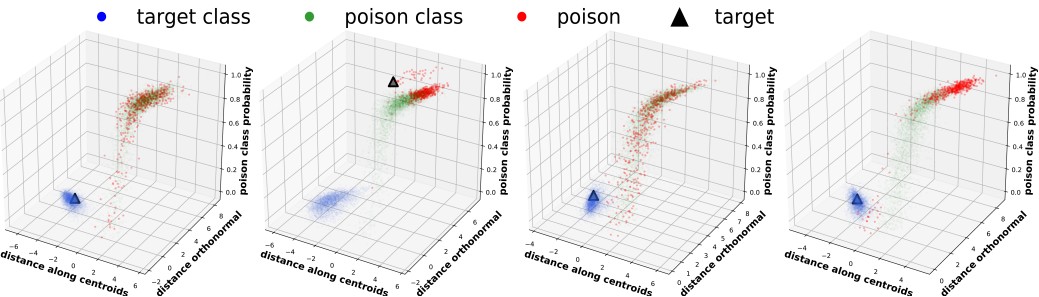

(a) **Undefended model**, clean (left) and retrained on poisoned
data containing gradient matching attacks (right).

(b) **Defended model**, clean (left) and retrained on poisoned
data containing gradient matching attacks (right).

Figure 8: 3D Visualization of the effects of a gradient matching attack (**Witches' Brew**) against an undefended and a defended model. The defended model significantly hinders feature collisions. The target image is marked by a black triangle and is originally part of the class marked in blue. The poisoned images are marked in red and are part of the class marked in green.

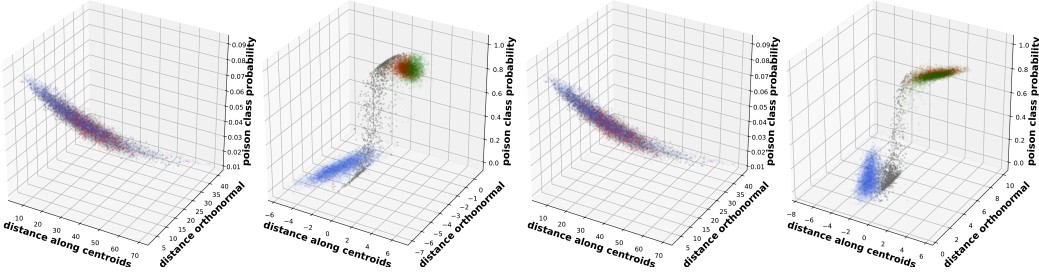

(a) **Undefended model**, clean (left) and retrained on poisoned
data containing backdoor patches. (right).

(b) **Defended model**, clean (left) and retrained on poisoned
data containing backdoor patches (right).

Figure 9: 3D Visualization of the effects of a **backdoor trigger** patch attack against an undefended and a defended model. The target trigger is applied to a number of target images shown in black and is originally part of the class marked in blue. The poisoned images are marked in red and are part of the class marked in green. Note how the black datapoints are associated with the poison class in the undefended case, but correctly associate with the target class in the defended case.

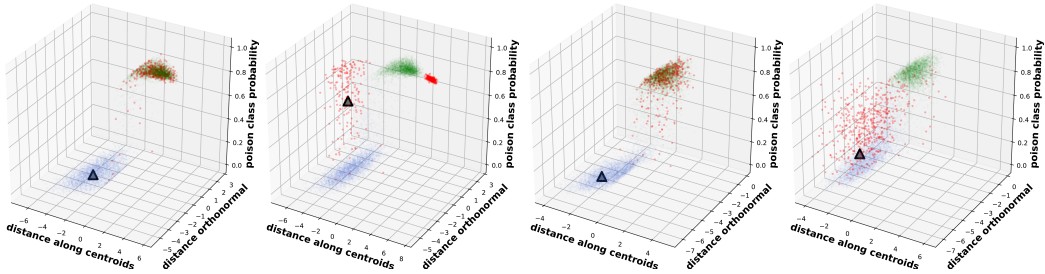

(a) **Undefended model**, clean (left) and fine-tuned on poisoned
data containing gradient matching attacks (right).

(b) **Defended model**, clean (left) and fine-tuned on poisoned
data containing gradient matching attacks (right).

Figure 10: 3D Visualization of the effects of a gradient matching attack (**Witches' Brew** with squared loss) against an undefended and a defended model. The defended model significantly hinders feature collisions. The target image is marked by a black triangle and is originally part of the class marked in blue. The poisoned images are marked in red and are part of the class marked in green.
We see that the attack effectively moves the decision boundary opposite of the target in feature space. However this is completely prevented in the defended model.

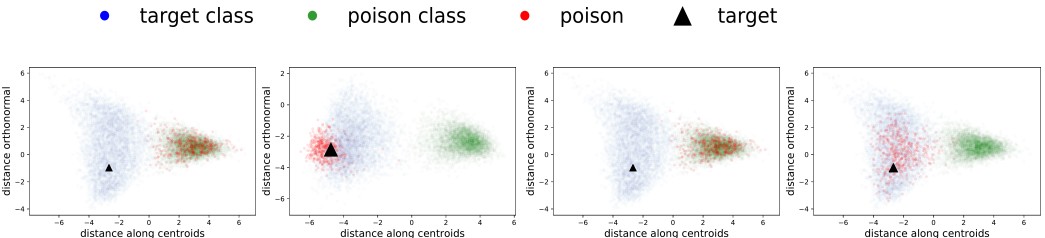

(a) **Undefended model**, clean (left) and fine-tuned on poisoned
data containing feature collisions. (right).

(b) **Defended model**, clean (left) and retrained on poisoned
data containing gradient matching attacks (right).

Figure 11: 2D Visualization of a feature collision attack (via **Poison-Frogs**) against an undefended and a defended model. The defended model significantly hinders feature collisions. The target image is marked by a black triangle and is originally part of the class marked in blue. The poisoned images are marked in red and are part of the class marked in green.

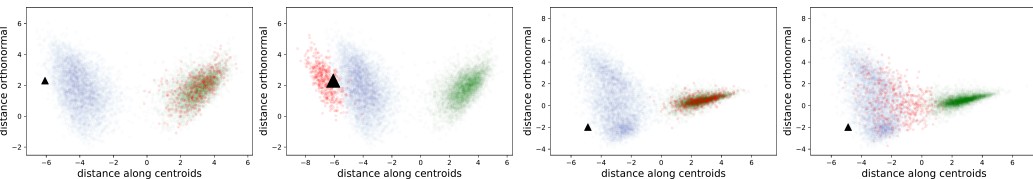

(a) **Undefended model**, clean (left) and fine-tuned on poisoned data containing feature collisions. (right).

(b) **Defended model**, clean (left) and retrained on poisoned data containing gradient matching attacks (right).

Figure 12: 2D Visualization of the effects of a feature collision (**Bullseye Polytope**) attack against an undefended and a defended model. The defended model significantly hinders feature collisions. The target image is marked by a black triangle and is originally part of the class marked in blue. The poisoned images are marked in red and are part of the class marked in green. Notably the strong collision seen in the baseline is inhibited by the defense.

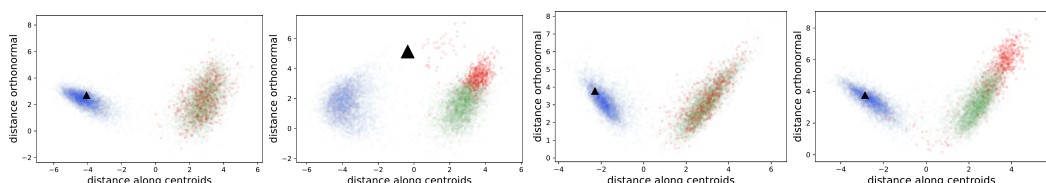

(a) **Undefended model**, clean (left) and retrained on poisoned data containing gradient matching attacks. (right).

(b) **Defended model**, clean (left) and retrained on poisoned data containing gradient matching attacks (right).

Figure 13: 2D Visualization of the effects of a gradient matching attack (**Witches' Brew**) against an undefended and a defended model. The defended model significantly hinders feature collisions. The target image is marked by a black triangle and is originally part of the class marked in blue. The poisoned images are marked in red and are part of the class marked in green.

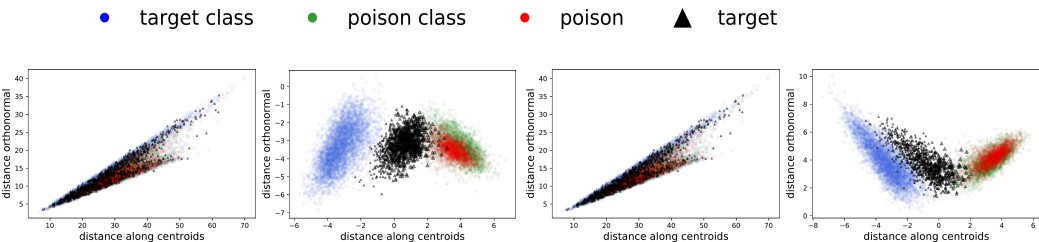

(a) **Undefended model**, clean (left) and retrained on poisoned data containing backdoor patches. (right).

(b) **Defended model**, clean (left) and retrained on poisoned data containing backdoor patches (right).

Figure 14: 2D Visualization of the effects of a **backdoor trigger** patch attack against an undefended and a defended model. The target trigger is applied to a number of target images shown in black and is originally part of the class marked in blue. The poisoned images are marked in red and are part of the class marked in green. Note how the black datapoints are associated with the poison class in the undefended case, but correctly associate with the target class in the defended case.

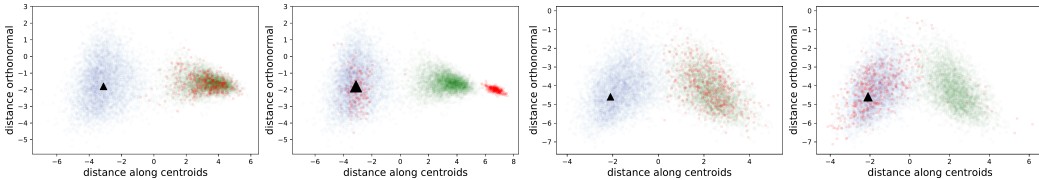

(a) **Undefended model**, clean (left) and fine-tuned on poisoned
data containing gradient matching attacks. (right).

(b) **Defended model**, clean (left) and fine-tuned on poisoned
data containing gradient matching attacks (right).

Figure 15: 2D Visualization of the effects of a gradient matching attack (**Witches' Brew** with squared error) against an undefended and a defended model. The defended model significantly hinders feature collisions. The target image is marked by a black triangle and is originally part of the class marked in blue. The poisoned images are marked in red and are part of the class marked in green.
We see that the attack effectively moves the decision boundary by placing poisoned data opposite of the target in feature space. However this is completely prevented in the defended model.

Table 6: Defenses against feature collision via (Aghakhani et al., 2020) in the transfer setting for a budget of $1\%$ and bound of $\varepsilon = 16$.

| Defense | Poison Success | Val. Acc. | Time |
|---|---|---|---|
| None | 100.00% ($\pm 5.59$) | 91.97% | 0:08:04 |
| Random Noise | 90.00% ($\pm 6.71$) | 90.45% | 0:08:33 |
| Deep K-NN | 75.00% ($\pm 9.68$) | 91.94% | 3:29:13 |
| Activation Clustering | 0.00% ($\pm 5.59$) | 91.34% | 0:07:08 |
| Spectral Signatures | 0.00% ($\pm 5.59$) | 92.09% | 0:09:10 |
| Diff. Priv. SGD ($n = 0.0001$) | 100.00% ($\pm 5.59$) | 92.72% | 0:08:08 |
| Diff. Priv. SGD ($n = 0.001$) | 100.00% ($\pm 5.59$) | 91.26% | 0:08:07 |
| Diff. Priv. SGD ($n = 0.01$) | 85.00% ($\pm 7.98$) | 69.78% | 0:08:12 |
| Adv. Training ($\varepsilon = 8$) | 5.00% ($\pm 4.87$) | 77.80% | 0:42:20 |
| Adv. Training ($\varepsilon = 16$) | 5.00% ($\pm 4.87$) | 58.98% | 0:42:37 |
| Adv. Poisoning ($s = 0.25$) | 90.00% ($\pm 6.71$) | 91.24% | 0:30:44 |
| Adv. Poisoning ($s = 0.5$) | 70.00% ($\pm 10.25$) | 89.83% | 0:35:06 |
| Adv. Poisoning ($s = 0.75$) | 10.00% ($\pm 6.71$) | 88.64% | 0:39:53 |

and due to the limitation that only the last layer is re-trained - leaving less control over the model. We record results in Table 6, finding that poison immunity can be effective even in a scenario that favors filter defenses, matching filter defenses, while beating adversarial training significantly in the trade-off against validation performance.

# E    EXPERIMENTAL SETUP

In general terms, the goal of our experimental setup is to standardize the experimental conditions encountered in various works in data poisoning to a degree that allows for convenient comparisons across attacks. Furthermore previous works have focused on showcasing the smallest possible adversarial modifications that still achieve a substantially malicious effect. However, such setups "on the edge" are broken too easily by any defenses, so that we generally consider stronger attacks in this work than in their original implementations. On the other hand, there is an upper limit to this design because attacks have to be sufficiently realistic, modifying only parts of the dataset within limits - unlimited adversarial modifications would allow for unlimited attack strength.

For all experiments shown in this work, we standardize the machine learning model that is attacked to a deep neural network for image classification, namely always a ResNet-18 model trained on the CIFAR-10 dataset. The ResNet-18 model follows (He et al., 2015), with the customary CIFAR-10 modification of replacing the initial 7x7 convolution and max-pooling with a 3x3 convolution. We train this model using SGD with Nesterov momentum ($m = 0.9$) with a batch size of 128 for 40

epochs with an initial learning rate of $0.1$, which is reduced by a factor of 10 after $\frac{3}{8}$, $\frac{5}{8}$ and $\frac{7}{8}$ of all epochs. The model is additionally regularized by weight decay with weight $5 \times 10^{-4}$. The CIFAR-10 dataset is augmented with horizontal flips and continuous random crops from images with a zero padding of 4.

We directly train with these hyperparameters in the FROM-SCRATCH setting. For the TRANSFER experiments we first train a ResNet-18 with this setup, which we call the base model, and then freeze its feature representation. We then retrain the linear layer from a random initialization with the same hyperparameters. For the FINE-TUNING experiments we drop the learning rate by $0.001$ before fine-tuning from the base model, likewise reinitializing the linear layer, but not freezing the feature representation. This setup for fine-tuning and transfer is arguably easier to attack than a transfer to an unknown dataset (as investigated for example in (Shafahi et al., 2018)), as all features are already optimized to be relevant to the given task, and as such we do not recommend it as the only evaluation of an attack, but believe it is an appropriate worst-case setting for the defense experiments considered in this work. Note that we apply a slightly different setting for the convex polytope attack of Zhu et al. (2019) which we will detail together with the attack in the next section and mark by TRANSFER* in the main table.

### E.1 MEASURING POISON EFFECTIVENESS

We run data poisoning attacks with the goal of maliciously classifying a single target image for all targeted data poisoning attacks, and with the goal of maliciously classifying 1000 images patched with a single target trigger for backdoor attacks. In both cases these images are drawn from the validation set without replacement. We report success for an attack if the target image, or patched image is classified with the adversarial label, we do not count mis-classifications into a third label. In all experiments, we then report *avg. poison success*. This metric represents the average success over $N$ random trials, where each trial consists of a randomly drawn target trigger or image, randomly chosen adversarial class, randomly chosen subset of images to be poisoned (from the adversarial class) and random model initializations. We control these trials with by specifying their random seed. All experiments in this work are based on the same 20 fixed trials, which we list by their seed within the supplemented code submission and as such comparable.

For all attacks we consider an $\ell^\infty$ bound of $\varepsilon = 16$ for targeted data poisoning attacks.

### E.2 ATTACK SETTINGS FOR $\ell^\infty$ THREAT MODELS

For all attacks we optimize the adversarial perturbation through projected descent (PGD). We found signed Adam with a step size of $0.1$ to be a robust first-order optimization tool to this end which we run for 240 steps, reducing the step size by a factor of 10 after $\frac{3}{8}$, $\frac{5}{8}$ and $\frac{7}{8}$ of total steps. Basic data poisoning attacks are often brittle when encountering simple data augmentations (Schwarzschild et al., 2020). As we include data augmentations in our experimental setup, we also include them during the attack algorithm for all iterative attacks, sampling a random augmentation in every update step and differentiating through the resulting transformation via grid sampling (Jaderberg et al., 2015). Some works such as Geiping et al. (2021) consider multiple restarts of the attack algorithm, however we consistently run all attacks with a single restart, mostly due to computational constraints, and given that restarts appear to confer only a minor benefit.

**Poison Frogs.** We implement the feature collision objective as proposed in Shafahi et al. (2018). However, while perturbation bounds were weakly enforced in the original version by an additional penalty, we instead optimize the objective directly by projected (signed) gradient descent in line with other attacks. We find this to be at least equally effective.

**Convex Polytope.** Poisoned data created by Convex Polytope can be brittle in terms of the amount of training data and optimizer settings (Schwarzschild et al., 2020). Therefore, to get an idea of how well our defense works against this attack, we implement a modified setting wherein the attack succeeds. This includes using a feature extractor trained on CIFAR-100, and training the last linear layer for only 10 epochs using the Adam optimizer with lr= $0.1$. We otherwise applied the attack as proposed in Zhu et al. (2019). This experiment is found in Table 7.

Table 7: Transfer* refers to the explicit setting of Zhu et al. (2019). The proposed defense significantly decreases success rates, even in this setting.

| Attack | Scenario | Undefended | Defended |
|---|---|---|---|
| Convex Polytope | Transfer* | 90.00% ($\pm 10.00$) | 40.00 % ($\pm 16.32$) |

**Bullseye Polytope.**   We directly re-implement the attack based on eq.(2) in Aghakhani et al. (2020).

**Witches' Brew.**   We implement gradient matching as in Geiping et al. (2021). However, we modify the original attack in the transfer setting. The original attack is posed for from-scratch attacks on large models and the objective of cosine similarity of parameter gradients does not scale well to small models. For small models (such as the transfer case, where only the last linear layer is retrained), we instead measure similarity in the squared Euclidean norm. We refer to this variant as gradient matching with squared error, e.g. Gradient Matching (SE) in table 1.

**MetaPoison**   We download premade poisoned datasets for MetaPoison from `https://github.com/wronnyhuang/metapoison`, validate their effectiveness and and then deploy our proposed defense using our own surrogate attack at the batch level, which we unroll for 2 steps as also proposed for the attacks in Huang et al. (2020). We defend using only the current estimate of the model as discussed in previous sections (instead of replicating the ensemble of 24 models used to create the poisoned dataset in some fashion). We train the usual ResNet-18 model on the downloaded poisoned CIFAR-10 for 40 epochs without data augmentations, conforming to the training setup without augmentations based on which the poisons were created. We download poisoned datasets for a budget of $1\%$ for the bird-dog setting with bird target ids $0$ to $9$ with perturbations bounded by $\varepsilon = 8$ in $\ell^\infty$-norm and perturbed by $4\%$ in color space.

**Hidden Trigger Backdoor**   For the hidden trigger backdoor attack of Saha et al. (2020) we use triggers identical to the ones used in the original work (see `https://github.com/UMBCvision/Hidden-Trigger-Backdoor-Attacks`). The number of poisons we use is indicated in the main body experiments, and is on same order as the number used in the original work. Specifically, we evaluate the attack on 1000 patched target images and choose a budget of $5\%$. The adversarial perturbations to the subset of poisoned data are optimized as described in the general attack settings, minimizing the hidden trigger objective of matching poison features to features of patched images.

## E.3   DEFENSE SETTINGS

**Input Noise**   As a sanity check we include a comparison to input noise. We draw random noise from the boundary of the set of allowed perturbations by independently sampling from a Bernoulli distribution for each value, and assigning either $-\varepsilon$ or $\varepsilon$ to each value.

**CutMix and Maxup**   We apply CutMix (Yun et al., 2019) as a defense against data poisoning as proposed in Borgnia et al. (2020). We attack this defense adaptively by creating poisoned data based on a clean model trained with CutMix as well. The same considerations apply for Maxup (Gong et al., 2020). We use Cutout (DeVries & Taylor, 2017) as a base augmentation for Maxup, and select the worst-case augmentation from four examples.

**Spectral Signatures**   We implement the defense as proposed in Tran et al. (2018b), using the provided overestimation factor of 1.5. We supply the attack budget as additional info for this defense.

**Deep K-NN**   We implement the defense as proposed in Peri et al. (2020), using the provided overestimation factor of 2. We supply the attack budget as additional info for this defense.

**Activation Clustering**   We run the defense of Chen et al. (2019), clustering the available training data into two clusters.

**Differentially private SGD** We implement a variant of differentially private SGD with gradient clipping to a value of 1 on a mini-batch level (as suggested in Hong et al. (2020)), and varying levels of Gaussian noise applied to the mini-batch gradient. Attacks can adapt to this defense by adding gradient noise to their surrogate estimation of gradients (this is mostly relevant for gradient matching where surrogate gradients appear explicitly).

**Adversarial Training** We implement straightforward adversarial training, starting from a randomly initialized perturbation and maximizing cross entropy for 5 steps via signed descent. Interestingly, for small $\varepsilon$ values, this defense can be overcome by the poisoner by creating poisoned data while adversarial noise is sampled and added during the poison optimization. This however this only helpful when the adversarial training $\varepsilon$ is smaller than the attack $\varepsilon$.

**Poison Immunity** For all implementations of adversarial poisoning we replicate the original objective of the attack in the mini-batch setting, but optimize for only 5 steps, based on features or gradients from the current model. The surrogate attacks are optimized via signed Adam descent with the same parameters as described in the attack section. For the hidden trigger backdoor attack we draw random patches as surrogate targets, and then optimize the $\ell^\infty$ perturbations for 5 steps as usual.

### E.4 ATTACK SETTINGS FOR $\ell^0$ THREAT MODELS

For backdoor triggers, we allow triggers with a size of 4 by 4 pixels, i.e. a rectangular arrangement of the $\ell^0$ bound of $\varepsilon = 16$. We allow a budget of $5\%$ of the dataset to be modified. We then imprint this patch in the lower right corner of all images in the poison set. During evaluation we imprint the same patch in the same location for 1000 target images. We first evaluate a "noisy checkerboard" patch, which is computed by sampling a Bernoulli variable for each patch pixel and RGB channel independently and assigning either 0 or 255. This patch is arguably not part of the distribution of CIFAR-10 images. Secondly, we select a resized firefox logo as the second patch, which leads to a patch that appears semantically similar to CIFAR-10 content.

We evaluate several possible defenses:

**Poison Immunity - Noise Patch** We sample a random noisy checkerboard pattern (Bernoulli samples in each pixel and channel as above) with a random rectangular shape with lengths within $[3, 12]$ for an approximate $\ell^0 < 45$ (overestimating the actual $\ell^0$ bound for a gray-box setting) as well as a random location. We sample such a patch from for every class in the dataset and then apply them to randomly chosen pairs of classes, replicating the attack without knowing the targeted class.

**Poison Immunity - Large Noise Patch** We repeat the previous setup with random noisy checkerboards sampled with lengths within $[8, 28]$, but otherwise the same setup as above. Note that even these large patches improve upon the patches with lengths in $[3, 12]$ only in minor ways for the semantic patch.

**Poison Immunity - Optimized Patch** Previously we only sampled random patches during the defense. This mirrors the methodology of previous sections on targeted attacks, in the sense that a data poisoning attack is used to attack each mini-batch in the same way that it would be used by an attacker - and the backdoor trigger attack also samples these patches randomly as described above. However, this is arguably a non-optimal approximation of the objective described in Equation (2) in the main body and further we can conjecture a more optimal backdoor trigger attack that would optimize patch contents. As such we optimize the contents of a patch with random rectangular shape as described for the noise patch, while still drawing the location at random. However, we need to choose a surrogate objective on which to optimize the patch. For this task we choose gradient matching and as such optimize the patch so that the gradients of patched target data and patched poisoned data are optimally aligned. We apply the same procedure as otherwise in this work and optimize with signed Adam for 5 steps with step size $\tau = 0.01$.

As we find in Table 2 though, this approach does not improve upon the noise patch sampling (and even performs worse). Possible reasons for this behavior are the choice of surrogate objective and optimization setting or the question of whether the patch location and shape should also be optimized, which would however further complicate the optimization.

**Poison Immunity - Image Patch** In contrast to sampling a noise patch, we can also sample this patch from in-distribution data, i.e. CIFAR-10 content. To do so we sample these patches from other images in the mini-batch. We choose a larger, but fixed size of $16 \times 16$ patches with a random location. To correct for the semantic impact of these patches on unrelated images we have to adjust the label information to match, so that finally, this option reduces exactly to CutMix (Yun et al., 2019), also proposed as a defense in Borgnia et al. (2021).

**Filtering Defenses** We apply activation clustering, spectral signatures and deep KNN as described for targeted attacks above.

**Differentially private SGD** We apply differentially private SGD as described for targeted attacks above with a noise level of 0.01 and a gradient clipping to 0.1.

### E.5 COMPUTATIONAL REQUIREMENTS

We run all reported experiments using `Nvidia GEFORCE RTX 2080 Ti` GPUs, using one GPU per experimental run which are scheduled from an internal SLURM system. We also ran select ablation pre-trials using a `Nvidia V100` setup. Training a robust model in the (most expensive) from-scratch setting requires approx. 4 hours and 10 minutes for the evaluated ResNet-18 on CIFAR-10 for the `RTX 2080 Ti`. After training a clean model (for the adaptive attack), the attack takes approx. 5 minutes. Finally the model is trained again on the poisoned data and its resilience against the poisoning is measured. In total this requires approx 8h of compute per experiment, of which we run 20 experiments for every data point with random poison-target class pairs and samples as described above. Training time reduces to approx. $2 \times 1$h for the fine-tuning setting and $2 \times 35$ minutes for the transfer setting.

### E.6 ASSET LICENCES

We use CIFAR-10 data (Krizhevsky, 2009) as found at `https://www.cs.toronto.edu/~kriz/cifar.html`. The code submission is based on forked repositories of the projects of Geiping et al. (2021) and Schwarzschild et al. (2020) as described in detail in the attached code submission folder.

