# OpenReview forum: "What Doesn't Kill You Makes You Robust(er): How to Adversarially Train against Data Poisoning"
_ICLR.cc/2022/Conference — ICLR 2022 Submitted_

### Official Review · Reviewer_dQnr · 2021-10-24

**Correctness:** 4
**Technical Novelty And Significance:** 3
**Empirical Novelty And Significance:** 3
**Recommendation:** 6
**Confidence:** 2

**Main Review:**

This paper is clear and easy to understand. The idea is interesting and the arguments are convincing. However, since it is some kind of combination of two existing robust models, I give weak acceptance based on the novelty. It would be great if the authors can provide more evidence on the novelty of this algorithm, e.g. what difficulties you met when designing this algorithm.

Below are some other minor comments:

(1) Could you provide some insights on why adversarial training performs worse than the proposed method in Figure 3? For example, what attack does adversarial training tend to defend, and what is the formula of the attack used in Figure 3?

(2) After adding adversarial training idea, the algorithm involves three levels of optimization. The adversarial training is known to be time consuming, so I expect the proposed algorithm takes an even longer time to run. Please report the running time of your experiment.

(3) It takes me several minutes to understand the layout of Figure 2: there are four groups (a) to (d) and each group contains two subfiguires. Please enhance the display of Figure 2. At the first glance I was thinking there are two groups (upper and lower), and the title of the upper group is "(a) Undefended model, clean (left) and fine-tuned on (b) Defended model, clean (left) and fine-tuned on".



**Summary Of The Paper:**

This paper propose to use the idea of adversarial training to defend data poisoning. The idea is interesting and the numerical results
also demonstrate the promising performance of this algorithm.

**Summary Of The Review:**

This paper is clear and easy to understand. The idea is interesting and the arguments are convincing. However, since it is some kind of combination of two existing robust models, I give weak acceptance based on the novelty.

---

> ### Author Response · Authors · 2021-11-18
> **Response to Reviewer dQnr**
>
> Thank you for your feedback for our work. We provide answers below, but let us know if you have further questions.
> > It would be great if the authors can provide more evidence on the novelty of this algorithm, e.g. what difficulties you met when designing this algorithm.
>
> We discuss some of these adaptations in the remark “Example: Defending against Gradient Matching” and in the context of adaptive attacks in Sec 3.2. We do want to highlight that the resulting defense that arises from Alg.1 is noticeably different from “only” adversarial training which we compare to in Figure 3.
>
>
> > (1)  Could you provide some insights on why adversarial training performs worse than the proposed method in Figure 3? For example, what attack does adversarial training tend to defend, and what is the formula of the attack used in Figure 3?
>
> Adversarial training does work to mitigate poison effectiveness but always at a significantly higher cost to model performance. This is unavoidable as the objective for adversarial training is Eq(1), where the worst-case loss is minimized. In contrast, for targeted poisoning, the arising objective is not constrained to minimize worst-case loss, but minimizes the worst-case poisoning effects of data points x_p on other data points x_t. This is an intermediate objective that reduces the effects of poisoned data points on unpoisoned data (compare also the visualizations in Fig. 2). While the optimal worst-case loss always induces a drop in model performance in the investigated scenarios, the optimal poison-robustness does provide a better trade-off (but we would not expect it to defend against test-time attacks where modifications on x_t directly affect x_t).
>
> > (2)  After adding adversarial training idea, the algorithm involves three levels of optimization. The adversarial training is known to be time consuming, so I expect the proposed algorithm takes an even longer time to run. Please report the running time of your experiment.
>
>  Thank you for bringing this to our attention. We now include run times in an extended version of Table 1 in the appendix.
>
>  > (3) Please enhance the display of Figure 2. At the first glance I was thinking there are two groups (upper and lower), and the title of the upper group is "(a) Undefended model, clean (left) and fine-tuned on (b) Defended model, clean (left) and fine-tuned on".
>
> We’re sorry for the confusion and now separate the subplots clearly with increased spacing.

---

> > ### Comment · Reviewer_dQnr · 2021-11-21
> > **Followup**
> >
> > Thanks for your clarification with a lot of useful experiment details.
> >
> > I'm still not convinced by the novelty issue. I think it is interesting to consider the topic in this paper, but the idea is not that novel so I keep weak accept now. In my point of view, the idea behind adversarial robustness is like "given a loss, we add some perturbation in the data to maximize the loss". As a result, the adversarial robustness and the robustness considered in this paper are similar in this aspect. Some evidence is needed to show that such an adaptation of attack in the problem in this paper is not trivial. If the adaptation causes severe problem and you further do modifications so that the adaptation finally works, then that would be really novel.

---

### Official Review · Reviewer_pSWq · 2021-10-31

**Correctness:** 3
**Technical Novelty And Significance:** 4
**Empirical Novelty And Significance:** 4
**Recommendation:** 8
**Confidence:** 3

**Main Review:**

Strengthes:

1. The proposed training objective for poison immunity is reasonable and well-motivated.
2. Experiments are thorough. Many baselines are compared. Results are promising.

Questions/Weaknesses:

1. Is the proposed defense objective in Eq. (4) a natural generalization of the original adversarial training objective in Eq. (1). In other words, is Eq. (1) a special case of Eq. (4)? The authors are suggested to discuss this. If it is not the case, the wording of some parts of the paper, such as the title of Section 3 (i.e., "Generalizing Adversarial Training to Data Poisoning"), may need to be modified to avoid misunderstandings.
2. There are some closely related papers [1, 2, 3], which the authors may want to discuss. In particular, [1] proposed "an adaptive defense", where they can teach a model to correctly classify both unperturbed and perturbed pictures generated by Fawkes and LowKey. Their procedure shares some similarities with this paper. [2] also extended the use of adversarial training for defending against one type of poisoning attacks (they called delusive attacks). Note that those papers are largely concurrent to this paper, so they would not affect the contribution of this paper. Yet, elaborating on the differences from those papers would increase the integrity of this paper.
3. A minor typo. A punctuation mark is missing after "those of clean data" on the third-to-last line on the second page.

[1] Radiya-Dixit et al., Data Poisoning Won’t Save You From Facial Recognition. ICML Workshop, 2021.
[2] Tao et al., Better Safe Than Sorry: Preventing Delusive Adversaries with Adversarial Training. NeurIPS 2021.
[3] Li et al., Anti-Backdoor Learning: Training Clean Models on Poisoned Data. NeurIPS 2021.

**Summary Of The Paper:**

This paper proposes a defense against backdoor attacks and targeted attacks. Specifically, the proposed defense injects randomly chosen poisoned instances and targeted instances into the training data, and then forces the model to have right behaviors on these instances. Experimental results demonstrate the effectiveness of the proposed method.

**Summary Of The Review:**

Overall, I am leaning towards acceptance. The proposed defense outperforms many existing defense strategies including differentially private SGD, adversarial training, various data augmentations, and filter defenses, providing a strong trade-off of robustness and accuracy.

---

> ### Author Response · Authors · 2021-11-18
> **Response to Reviewer pSWq**
>
> Thank you for providing us with this feedback. We answer these questions below, but let us know if you have further questions:
>
> > Is the proposed defense objective in Eq. (4) a natural generalization of the original adversarial training objective in Eq. (1). In other words, is Eq. (1) a special case of Eq. (4)? The authors are suggested to discuss this. If it is not the case, the wording of some parts of the paper, such as the title of Section 3 (i.e., "Generalizing Adversarial Training to Data Poisoning"), may need to be modified to avoid misunderstandings.
>
> The objective in Eq(4) can technically be seen as a generalization of adversarial training Eq.(1) in the setting where we move from targeted data poisoning (i.e. model integrity attacks) to untargeted data poisoning (i.e. model availability). This can be done by choosing x_t = x_p and choosing y_t so that the loss over x_t is maximized instead of minimized, simplifying the outer objective in Eq.(2) to max x_p L(x_p, y_p, theta(x_p)).
> This is arguably not that straightforward, so we have adapted our text to clarify the relationship to adversarial training.
>
> > There are some closely related papers [1, 2, 3], which the authors may want to discuss. In particular, [1] proposed "an adaptive defense", where they can teach a model to correctly classify both unperturbed and perturbed pictures generated by Fawkes and LowKey. Their procedure shares some similarities with this paper. [2] also extended the use of adversarial training for defending against one type of poisoning attacks (they called delusive attacks). Note that those papers are largely concurrent to this paper, so they would not affect the contribution of this paper. Yet, elaborating on the differences from those papers would increase the integrity of this paper.
>
> Thank you for mentioning these references. We have included them and the following description in our updated manuscript. Summarizing the relationship of this work to these contemporary works in brief:
> *  [1], Radiya-Dixit and Tramèr, provide reasons why attacks against model availability are naturally at a disadvantage as the defender (who goes second in the adversarial game) can notice model degradation and adapt their training scheme by using different models or training schemes. For model integrity attacks that we investigate in this work, this defense does not work as well, as there exists no method to robustly detect the breach of model integrity.
> * [2] Tao et al., provide adversarial training defenses against model availability (also called delusive poisoning). In this untargeted scenario, Eq(4) reduces to Eq(1) as described above, so that direct adversarial training is already a good defense against model availability attacks.
> * [3]  Li et al interestingly come up with a different approximation to Eq(2). Their approximation requires successful detection of poisoned samples, which does not always work well, but if it does, allows for a simplified training where the model can be directly minimized on clean samples and maximized on detected poisons (making it unnecessary to generate poisons on-the-fly). This works especially well for patch attacks (l^0) where generation on the fly is difficult, but detection is more possible than for the l^infty attacks that we focus on.
>
> > A minor typo. A punctuation mark is missing after "those of clean data" on the third-to-last line on the second page.
>
> We have also adresses this.

---

> > ### Comment · Reviewer_pSWq · 2021-11-19
> > **Follow-up**
> >
> > Thank you for your efforts in addressing my comments and the others. Mostly I'm satisfied.
> >
> > > [1], Radiya-Dixit and Tramèr, provide reasons why attacks against model availability are naturally at a disadvantage as the defender (who goes second in the adversarial game) can notice model degradation and adapt their training scheme by using different models or training schemes. For model integrity attacks that we investigate in this work, this defense does not work as well, as there exists no method to robustly detect the breach of model integrity.
> >
> > The above response may be *wrong* about Radiya-Dixit and Tramèr [1]. In particular, their defense has similar goals to this work. As far as I can tell, they are studying particularly two *integrity* attacks including Fawkes and LowKey. In Fawkes' paper, we can see that "Fawkes’goals are to mislead rather than frustrate", which means that it's indeed an integrity attack, and LowKey has the same goal as Fawkes.
> >
> > Therefore, the defense in [1] actually has the potential to resist the model integrity attacks investigated in this work.

---

> > > ### Author Response · Authors · 2021-11-19
> > > **Follow-up Clarification**
> > >
> > > Thanks for the additional clarification.  Fawkes is indeed an integrity attack by design. LowKey considers poisoning gallery images of pre-trained face recognition systems, but we agree with you that our defense may also be applied to defending against these attacks and that the defense recommendations from [1] could be useful against targeted data poisoning and backdoor attacks.
> > >
> > > We have amended our draft to rectify our characterization of [1], and we appreciate your feedback.

---

> > > > ### Comment · Reviewer_pSWq · 2021-11-20
> > > > **Thanks**
> > > >
> > > > Thanks for the prompt clarification. Based on the response and the revision, I would like to increase my score to accept.

---

### Official Review · Reviewer_E8z9 · 2021-10-31

**Correctness:** 2
**Technical Novelty And Significance:** 2
**Empirical Novelty And Significance:** 2
**Recommendation:** 3
**Confidence:** 4

**Main Review:**

It is interesting to extend adversarial training to defending against data poisoning attacks. I reviewed this paper before and the following problems have not yet been addressed.

1. The key of the proposed training is based on the expectation that the generated poisoned data can counter the effect from the real injected data. However, this relies how many data are polluted in the training set. If only a very small portion is injected, then it is possible that the proposed training may eliminate the effect from poisoned data. What if there is a much larger portion of the training set poisoned, say 50% or even 80%? Can the proposed training still reduce the attack success rate without sacrificing normal accuracy? For those poisoning attacks that change labels of poisoned data, the chosen label $y_t$ in Algorithm 1 may still be the true target label, which does not counter the original effect of poisoned data. The current version of the paper does not seem to study the impact of different poisoning rates on the proposed approach.

2. The results show the proposed training is effective against $L_\infty$ but not against $L_0$ (in Appendix D.4). This may be due to the approximation of generating trigger patterns during training. The paper only uses 5 steps to generate those patterns. This is useful for $L_\infty$ attacks as it is more like classical adversarial training. As long as the perturbation is pointing towards the target direction, the added perturbation can help the model to be insensitive to those types of perturbations. However, for patch-like attacks, a small number of steps may just generate some random pattern and do not help the training. The results on the semantic patch is poor by the proposed approach. The best result is achieved using image patch, which is actually the baseline CutMix. This means the proposed method is no better than the baseline. This limitation should be discussed early in the paper, preferably in the introduction. Or the authors can adjust their threat model and just focus on $L_\infty$ attacks. The introduction of the current version does not clarify the limitation of the proposed method regarding the threat model. The evaluation on $L_0$ attacks does not include more studies.

3. Generating $L_\infty$ poisoned data during the proposed training procedure is similar to crafting universal perturbations. How about using universal adversarial training [1] to defend against data poisoning attacks? This is not investigated in the current version.

4. The evaluation is only conducted on one model (ResNet-18) and one dataset (CIFAR-10). The observations and experimental results may not be general and applicable to other cases. Including more model structures (e.g., VGG, Inception, etc.) and datasets (GTSRB, ImageNet or its subset, etc.) can provide a better understanding of the performance of the proposed training method. There is a public dataset of poisoned models called TrojAI dataset. The original training datasets and the code for generating those models are public available. An extensive evaluation on a larger benchmark can better assess the performance of the proposed approach.

5. There are missing critical results. The accuracies on clean data are missing in Table 1 and Table 2.

6. The writing of the paper needs improvement. This paper categorizes backdoor attacks into backdoor trigger attacks and targeted data poisoning, which is good. But then the paper lists 5 attacks that are evaluated in the experiment. The two parts are not well organized. I would suggest to put the relevant attacks in the corresponding categories and then elaborate their similarities and differences. Subsections can be used to highlight different parts of the related work. For instance, use subtitle "poisoning attacks" to indicate the discussion on attacks and "poisoning defenses" to indicate the discussion on defenses. The references to figures and tables are not consistent. The paper uses "Figure 2a" on page 6 but uses "fig. 2c" on page 7. The paper directs the readers to the supplementary material on page 7 but does not point out the specific section.

[1] Shafahi, Ali, et al. "Universal adversarial training." Proceedings of the AAAI Conference on Artificial Intelligence. Vol. 34. No. 04. 2020.

**Summary Of The Paper:**

This paper proposes to leverage classical adversarial training to defend against data poisoning attacks. Specifically, the paper splits each training batch randomly into two sets: one set for generating the trigger pattern and the other for injecting the pattern. The subject model is then trained on the modified batch. This paper evaluates the proposed training method on a few $L_\infty$ based backdoor attacks and shows that it has a better balance between normal accuracy and defense performance compared to baselines.

**Summary Of The Review:**

1. There is no study on the impact of different poisoning rates.
2. The method is only effective against $L_\infty$ but not $L_0$ and it is not clarified in the introduction.
3. A baseline defense method is not evaluated in the paper.
4. The number of evaluated models and datasets is small and limited.
5. The writing needs improvement.

---

> ### Author Response · Authors · 2021-11-18
> **Response to Reviewer E8z9**
>
> Thank you for your extensive feedback on this work. We have now run extensive experiments to address each of your individual concerns, and we have updated our Appendix accordingly.  See below for a description of these experiments.
>
> We also want to stress that we addressed key parts of your previous review, especially regarding extended comparisons to adversarial training with low epsilon, structuring of l^infty vs l^0 settings, and presentation of this work.  We also stress that the TrojanAI benchmark is focused on sanitizing already poisoned models, whereas we are interested in robustly training new models from scratch, so the setup of this benchmark is inapplicable to us.
>
> We appreciate the additional feedback offered in this review and have taken all other comments to heart and addressed them as follows:
>
> > The key of the proposed training is based on the expectation that the generated poisoned data can counter the effect from the real injected data. However, this relies how many data are polluted in the training set
>
> Thanks for pointing out that the effects of increased poison budget were not well researched in our submission version. We now include a full study in the Appendix where we look at the gradient matching attack and scale the budget of the attack in the entire range of 1% to 10% (10% is the maximum possible budget for this attack on CIFAR-10 as all training samples from the poisoned class are poisoned). Even in these challenging situations (where 5000 data points are modified within eps=16 bounds to misclassify a single fixed target, the defense reduces poison effectiveness.
>
>
> > Generating  poisoned data during the proposed training procedure is similar to crafting universal perturbations. How about using universal adversarial training [1] to defend against data poisoning attacks? This is not investigated in the current version.
>
> We have now implemented universal adversarial training (UAP) as a defense and ran several sets of experiments with this defense which we also include in Figure 3. Our defense yields significantly better robustness than UAP for both eps=8 and eps=16.
>
> > The evaluation is only conducted on one model (ResNet-18) and one dataset (CIFAR-10). The observations and experimental results may not be general and applicable to other cases. Including more model structures (e.g., VGG, Inception, etc.) and datasets (GTSRB, ImageNet or its subset, etc.) can provide a better understanding of the performance of the proposed training method.
>
> Thanks for the suggestion.  We have now added additional results for other models trained on GTSRB to  the Appendix. Concretely we now also evaluate gradient matching attacks against GTSRB in the from-scratch setting as well as feature collision attacks (via bullseye polytope) against GTSRB in the transfer setting.
> We have run these experiments over the last few days, each containing multiple trials as described in our experimental setup.  We will continue to run additional combinations given enough time.  Are there any additional experiments in this vein that would further address your concerns?
>
> > There are missing critical results. The accuracies on clean data are missing in Table 1 and Table 2.
>
> Thanks for the suggestion.  We have now introduced large extended versions of Table 1 and 2 in the Appendix that contain both clean accuracies and run time measurements for each experiment in Table 1 and each experiment in Table 2.  We do want to point out that comparisons between clean accuracies across threat models is unclear, but we hope that this can give a better impression of the impact of the defense for each threat model.
>
> Furthermore we have implemented all writing suggestions and thank you for bringing them to our attention.

---

> > ### Comment · Reviewer_E8z9 · 2021-11-22
> > **Thanks for the Response**
> >
> > The results in Figure 5 validate my concern that as long as the attacker increases the poisoning rate to some extent, the proposed method is not able to defend. With 7% poisoned training data, the proposed method can only reduce the ASR to 50%.
> >
> > The current version still does not clarify the focus of this paper, which mainly defends against $L_\infty$ based attacks. As pointed out in the original review, the current results on $L_0$ are no better than the baseline. Unless more results can be provided to demonstrate the advantage of the proposed method over baselines on $L_0$, I would suggest to make it clear in the paper.
> >
> > Why the attack success rates for undefended models on GTSRB are so low (only 40% for Gradient Matching and 10% for Bullseye Polytope)? The ASRs on CIFAR have more than 80%. This implicitly supports my concern on experiments with limited number of model structures and datasets as generalizability is important.
> >
> > Thanks for improving the writing. There are some parts still requiring improvement. For instance, the normal accuracies should be incorporated in the original table instead of creating a new table in the appendix as those results are important to show that the defense does not have a large negative impact on normal functionalities.

---

> > > ### Author Response · Authors · 2021-11-22
> > > **Additional Responses**
> > >
> > > Thank you for the additional feedback. We'll answer these questions below:
> > >
> > > > The results in Figure 5 validate my concern that as long as the attacker increases the poisoning rate to some extent, the proposed method is not able to defend. With 7% poisoned training data, the proposed method can only reduce the ASR to 50%.
> > >
> > > The discussed attack was originally described in ranges from 0.1% to 1% of the dataset poisoned, corresponding to 1% to 10% of this class poisoned, based on assumption that the attacker might only be able to poison a small fraction of training data. Now 70% of all data points in the relevant class are poisoned, but the defense still reduces the ASR to 50%, we think this is a strong result! Note that this attack leads to misclassification of only a single target image. Now 3500 of 5000 training set images are modified to change the classification of this single target image, yet the defense continues to reduce attack success noticeably.
> > >
> > > > The current version still does not clarify the focus of this paper, which mainly defends against  based attacks.
> > >
> > > The proposed defense still works in the l^0 setting, it just reduces to an already known defense by CutMix. This happens because the optimization problem is best solved by sampling instead of optimization in this threat model. We'll work on clarifying this further.
> > >
> > > > Why the attack success rates for undefended models on GTSRB are so low (only 40% for Gradient Matching and 10% for Bullseye Polytope)? The ASRs on CIFAR have more than 80%.
> > >
> > > We will spend more time to figure out how to tune previous attacks to improve their performance on GTSRB (which is a simpler dataset than CIFAR10 and so has potentially less room for non-robust adversarial features). Yet, this is orthogonal to the strength of the defense that we propose in this work.

---

### Official Review · Reviewer_xUH6 · 2021-11-04

**Correctness:** 4
**Technical Novelty And Significance:** 3
**Empirical Novelty And Significance:** 2
**Recommendation:** 6
**Confidence:** 3

**Main Review:**

### Strengths
- The introduced method is sounds and a simple adaptation of adversarial training to make it more effective against data poisoning attacks.
- The paper is well-written and flows smoothly
- The authors put decent effort in detailing prior work and replicating their results within the same framework to foster reproducibility. The authors attach the code to the submission.
- The experimental setup is detailed in the appendix.

### Weaknesses
- The analysis section S4 and the corresponding qualitative results (Fig 2) are hard to follow and understand. What is the takeaway in this section? Making this clear in the paper/captions is important in my opinion.

### Small changes:
- Typo: “Defineda” -> “defined”
- Fig 2 captions are almost overlapping. Try to increase the space between the subplots for better readability.


**Summary Of The Paper:**

The paper adapts adversarial training to build robust models against data poisoning. Specifically, the paper shows how one can effectively apply data-poisoning attacks in the training loop to successfully defend against potential data poisons. The introduced method is rigorously evaluated by (adaptive) known poisoning attacks. The paper demonstrates that the new defense outperforms existing data-poisoning defenses on CIFAR-10. Finally, the paper qualitatively analyzes the feature space to understand the effect of the proposed defenses.

**Summary Of The Review:**

Overall, I believe the paper is good in that it introduces a simple yet effective adaptation of adversarial training to make it effective against data poisoning. The presented empirical results are convincing. Though, I don’t quite understand the qualitative assessment of the introduced defense, so it would be great if the authors can improve the presentation there.

---

> ### Author Response · Authors · 2021-11-18
> **Response to Reviewer xUH6**
>
> > The analysis section S4 and the corresponding qualitative results (Fig 2) are hard to follow and understand. What is the takeaway in this section? Making this clear in the paper/captions is important in my opinion.
>
> We thank the reviewer for their feedback. We have incorporated the suggested changes and edited our descriptions in Section 4. The changes can be found in the uploaded revision.
> We now highlight more strongly that the defense leads to qualitative change in the feature space of the defended models. This is best observed by comparing the spread of poisons (red) in each of the 4 subplots before and after the attack.
> Let us know if this helps to clarify the analysis.
>
> > Small changes:
>
> We have further incorporated these small changes, especially improving the spacing of Fig.2.

---

### Decision · Program_Chairs · 2022-01-20

**Decision:**

Reject

**Comment:**

Reviewer E8z9 remained with a number of serious concerns, including efficacy of the defense in higher poisoning setting, overclaiming contributions in terms of L0 defenses (which are mostly achieved by the baseline CutMix), as well as novelty and generalizability of the approach. The author responses were unconvincing, and all other reviewers participated in the discussion, conceding that they too were unable to provide compelling arguments against E8z9's comments. Other reviewers claimed that these drawbacks may be "acceptable" for a first step, but were not willing to defend it very strongly.

We note that E8z9 claims they are a reviewer for a previous version of the paper and these issues were present before. The authors claim that E8z9's key points had been addressed in this version, but the reviewer maintains that the issues still persist in the latest version. The authors are advised to take their comments into account for further versions of this paper.